# Bayesian Optimization of Machine Learning Classification of Resting-State EEG Microstates in Schizophrenia: A Proof-of-Concept Preliminary Study Based on Secondary Analysis

**DOI:** 10.3390/brainsci12111497

**Published:** 2022-11-04

**Authors:** Ahmadreza Keihani, Seyed Saman Sajadi, Mahsa Hasani, Fabio Ferrarelli

**Affiliations:** 1Department of Psychiatry, University of Pittsburgh, Pittsburgh, PA 15213, USA; 2Department of Medical Physics & Biomedical Engineering, School of Medicine, Tehran University of Medical Sciences, Tehran 1416634793, Iran; 3Institute of Medical Science and Technology, Shahid Beheshti University, Tehran 1985717443, Iran

**Keywords:** microstate analysis, schizophrenia, optimized machine learning, microstate map correlation, resting-state EEG

## Abstract

Resting-state electroencephalography (EEG) microstates reflect sub-second, quasi-stable states of brain activity. Several studies have reported alterations of microstate features in patients with schizophrenia (SZ). Based on these findings, it has been suggested that microstates may represent neurophysiological biomarkers for the classification of SZ. To explore this possibility, machine learning approaches can be employed. Bayesian optimization is a machine learning approach that selects the best-fitted machine learning model with tuned hyperparameters from existing models to improve the classification. In this proof-of-concept preliminary study based on secondary analysis, 20 microstate features were extracted from 14 SZ patients and 14 healthy controls’ EEG signals. These parameters were then ranked as predictors based on their importance, and an optimized machine learning approach was applied to evaluate the performance of the classification. SZ patients had altered microstate features compared to healthy controls. Furthermore, Bayesian optimization outperformed conventional multivariate analyses and showed the highest accuracy (90.93%), AUC (0.90), sensitivity (91.37%), and specificity (90.48%), with reliable results using just six microstate predictors. Altogether, in this proof-of-concept study, we showed that machine learning with Bayesian optimization can be utilized to characterize EEG microstate alterations and contribute to the classification of SZ patients.

## 1. Introduction

Schizophrenia (SZ) is a psychiatric disorder whose neurobiological underpinnings are still largely unknown. One of the most widely used techniques in SZ research is electroencephalography (EEG), which measures electrical activity in the brain. EEG signals reflect the oscillation and large-scale synchronization of underlying neural populations [1] and therefore can be used to investigate the synchrony and dynamics of neural circuits in SZ patients [2,3]. In patients with SZ, EEG abnormalities have been reported from studies measuring event-related potential paradigms, including face processing and mismatch negativity [4,5], sleep EEG characteristics [6,7], graph analysis measures of resting state EEG [8], and EEG power spectra, which are the most commonly computed features [9,10,11]. Statistical features, such as mean, skewness, and kurtosis were also investigated in SZ [12,13]. However, most of these EEG-based features have failed to take advantage of the EEG millisecond resolution [14,15,16,17,18].

Microstates are topographic maps of resting state brain activity that exploit the EEG temporal resolution. Specifically, Lehmann et al. first demonstrated the existence of microstates by segmenting EEG at the sub-second level, generating quasi-stable and evenly patterned data occurring at 80–120 ms intervals [19]. These images do not gradually merge or overlap in time; instead, a single map abruptly switches to another (i.e., microstates are quasi-stable periods of single maps) [20]. Microstate changes have been described as “atoms of cognition” [21] or “building blocks of mentation” [22]. For instance, distinct changes in EEG microstates have been associated with specific cognitive functions, sensory inputs [23], and tasks requiring reasoning [20]. Four main microstates (respectively Type A, Type B, Type C, and Type D) have been described, and the auditory, visual, default mode and dorsal attention pathways have been shown to relate to these microstates, respectively [24,25,26]. Several studies have reported microstate alterations in patients with SZ relative to healthy control subjects [14,15,21,27,28,29], and based on this increasing body of evidence, it was recently suggested that EEG microstates features could be used for SZ classification [27,30].

The heterogeneity and complexity of schizophrenia symptoms challenge an objective diagnosis; thus, a single predictor would likely be affected by such heterogeneity and may not reach the threshold to be detected by traditional statistical approaches. In contrast, more complex statistical models, such as those used in machine learning approaches, can be utilized to evaluate the combination of several predictors for the classification of SZ concurrently [31]. Although previous studies investigated SZ classification with machine-learning-based clustering algorithms [32,33,34], these studies have mostly used univariate or conventional machine learning methods, rather than optimized multivariate analyses [18]. Choosing optimized SZ classification is challenging and comprises multiple hyperparameters, which are set before the training process and define how the model can best fit the data [31,35]. There are two main types of hyperparameter selection methods: manual and automatic search. Manual search requires experienced users, whereas automatic methods, including grid and random searches, are more user independent [36]. Random search has solved the expensive cost of exhaustive searching in grid search and has proved to be more efficient in high-dimensional space, even though it may be unreliable for some complex models [36]. Choosing the best-fitted model with tuned hyperparameters is an optimization problem with black-box objective function, and several recent studies indicated that Bayesian optimization outperforms other optimization methods [37,38,39].

A few studies applied machine learning algorithms on SZ microstate predictors [27,30,40,41], but there is no study that evaluated the optimization multivariate analysis on these measures. In this study, we hypothesized that using optimized multivariate analysis along with numerous dependent and independent variables (i.e., microstate features), including newly computed microstate measures, would contribute to efficiently discriminating SZ patients from healthy subjects. Thus, we aimed to discriminate between EEG recordings of individuals diagnosed with SZ and healthy control participants using the Bayesian optimized machine learning approach and the best-fitted model based on the most important microstate features.

## 2. Materials and Methods

### 2.1. Dataset

We used a publicly available EEG dataset, which included 14 schizophrenia patients (7 males: 27.9 ± 3.3 years, 7 females: 28.3 ± 4.1 years) and 14 healthy controls (7 males: 26.8 ± 2.9, 7 females: 28.7 ± 3.4 years) from the Institute of Psychiatry and Neurology in Warsaw, Poland [42,43]. The patients met the ICD10 criteria for paranoid schizophrenia, according to the International Classification of Diseases (category F20.0). All participants were given a written description of the protocol and signed a consent form to participate in the study. A minimum age of 18 was required, as well as an ICD-10 diagnosis of F20.0 and at least a seven-day medication washout period was required before EEG was performed. The Warsaw Institute of Psychiatry and Neurology’s Ethics Committee approved the study protocol. Pregnancy, organic brain pathology, severe neurological diseases (e.g., epilepsy, Alzheimer’s disease, and Parkinson’s disease), the presence of a general medical condition, and the first episode of SZ, were all considered exclusion criteria. The 14 patients who completed the study were matched in gender and age to the control group. Fifteen minutes of resting-state EEG eye-closed were recorded from 19 channels with a sampling frequency of 250 Hz using the international 10/20 EEG system montage.

### 2.2. EEG Data Pre-Processing

We used EEGLAB [44] to pre-process the EEG dataset before performing microstate analyses. This step was completed based on well-established pre-processing methods [14,15,22,27,45]. Briefly, data were filtered between 2 and 20 Hz and re-referenced to the common average electrode. Then, EEG signals were segmented into 5-second nonoverlapping epochs. Outlier epochs were removed with a variance threshold (i.e., >3 standard deviation). Overall, 2511 artifact-free EEG epochs for schizophrenia patients and 2511 artifact-free epochs for healthy control subjects were utilized for the microstate analysis (Figure 1).

### 2.3. Microstate Analysis

The entire EEG signal can be represented by a small set of topographic maps that alternate at discrete intervals when EEG is viewed as a topography of electric potentials evolving over time [46]. These topographic maps are named microstates. These images/topographies do not gradually merge or overlap in time; instead, a single map dominates for roughly 80–120 milliseconds before abruptly switching to another map (i.e., microstates are quasi-stable periods of single maps) [19]. To extract the relevant maps, the root-mean-squared potential differences at all N electrodes (i.e., $V_{i}\left(t\right)$) from the mean of instantaneous potentials across electrodes (i.e., $V_{\mathrm{mean}}\left(t\right)$) were computed to characterize the EEG global field power (GFP) [47].
(1)$$\mathrm{GFP}\;=\sqrt{\frac{\sum_{i}^{N}{\left(V_{i}\left(t\right)-V_{\mathrm{mean}}\left(t\right)\right)}^{2}}{N}}$$

Topographies of the local maxima of the GFP curve were identified based on the above equation. We then used a modified K-means clustering approach and global explained variance (GEV) criteria [27,48,49] by setting the re-run parameter to 20 times, the convergence threshold at 10^−6^, and the maximum number of iterations to 1000 [33,44].

We identified four microstates (i.e., A, B, C, and D, Figure 2), which reflected the activity of the EEG channels with almost 70% of the total topographic variance, and labeled the topography at each GFP peak as one of these microstates. Five categories of features were computed for these microstates (Figure 3 and Appendix A): (1) the average number of times per second that each microstate occurred during the EEG recording (occurrence); (2) the average amount of time each microstate lasted after it occurred (duration); (3) the percentage of total recording time in which each microstate was dominant (coverage); [19] (4) the average GFP during microstate dominance (mean GFP) [28]; and (5) the average correlation between each labeled GFP peak map (i.e., A) and the corresponding microstate template (microstate map correlations) [34]. All microstate features are summarized in Table 1.

### 2.4. Predictors’ Ranking and Classification

We ranked the 20 features/predictors described above using chi-square tests (“fscchi2”) embedded in the Statistics and Machine Learning Toolbox of MATLAB software (MathWorks, Inc., Natick, MA, USA, 2020) (Figure 3). Then, an optimized machine-learning approach was applied to classify the combination of ranked microstate features for SZ classification by adding each feature incrementally. Of note, machine learning classification can be subject dependent or subject independent. In the subject-independent classification, the model is tested using unseen subject’s data, (i.e., the data that are used to test the model are not included in the training phase), while here in the subject-dependent classification, training and testing sets are randomly split, and EEG-segment data from the same subject are included in both sets [31]. Because our cohort of SCZ and HC subjects was rather small, subject-dependent classification was employed. In each run, 80% of the data was utilized in the training process with a 5-fold cross-validation method to avoid overfitting, and the remaining data (20%) were used to test for the best-fitted trained model. Histograms of subject-based features used for this study to show the within-group subject variabilities of microstate features were also computed (see Appendix A). Accuracy, area under the curve (AUC), sensitivity, and specificity were used to assess classifier performance. Furthermore, we used the leave-one-out validation method by excluding the data of one individual per group each time to examine the subject-independent design in addition to the current approach. The trained models on the remaining dataset were tested on these two individuals, who each time were left out. Our results are comparable in both designs (output measures in the range of about 90% for subject independent vs. subject dependent) in Appendix A. The classifier algorithms were selected from a pool of existing classifiers based on prior EEG studies that highlighted the important application role of ML as a real-time health monitoring system for stroke prognostics [50], access ischemic stroke-derived cortical impairment [51] and other biomedical engineering works [50,51,52,53,54,55]. The Shapley additive explanations (SHAP) or Shapley values of features were computed using the “shapley” function embedded in the Statistics and Machine Learning Toolbox of MATLAB2022b, which explains the deviation of the prediction for the query points from the average prediction, due to the feature (Figure 4). For each query point, the sum of the Shapley values for all features corresponds to the total deviation of the prediction from the average [56,57]. The details of the machine learning optimization analysis are described below.

### 2.5. Bayesian Optimization of Classification

To perform Bayesian optimization of machine learning classification, we employed an algorithm comprising 2 principal steps (Equations (S2) and (S3) in Appendix A), where ${\mathrm{DATA}}_{1:t-1}=\{x_{n},{\;y}_{n}\}\begin{array}{c} t-1\\ n=1 \end{array}$ defined the training dataset with the t − 1 observation of an unknown function (Appendix A).

To automatically choose the machine learning algorithm with tailored hyperparameters, the Statistics and Machine Learning Toolbox^TM^ (MATLAB and Release 2020b, The MathWorks, Inc., Natick, MA, USA) was utilized with the “fitcauto” function [37,38,56]. A multi-TreeBagger model of the objective function was included in the Bayesian optimization approach of “fitcauto”. The objective function of this model differed from the Gaussian process model implemented by other machine learning toolbox functions using Bayesian optimization, and the next point to be examined was determined by an acquisition function (i.e., expected improvement). The output of the “fitcauto” algorithm was the point with the lowest objective function value among the points assessed during the optimization. This method automatically chose the best machine learning method for training data from among the most applicable machine learning methods (e.g., discriminant analysis [‘discr’], ensemble learning [“ensemble”], kernel classifier [‘kernel’], k-nearest neighbor [‘knn’], support vector machine classifier (SVM), linear classifier [‘linear’], naive Bayes classifier [‘nb’], neural network classifier [‘net’], and decision tree classifier [‘tree’]). When the optimization process was completed, “fitcauto” returned the trained model for the entire train dataset to perform classification [57].

## 3. Results

Four microstates were identified: A, B, C, and D that exhibited right-frontal left-posterior, left-frontal right-posterior, midline frontal-occipital, and midline frontal topographies respectively. Microstate scalp topographies of patients diagnosed with SZ and healthy control subjects are shown in Figure 2. We also computed and compared twenty microstate features, including occurrence, duration, coverage, and microstate correlation maps between SZ patients and HC subjects. We found that most of the predictors were significantly different between groups after Bonferroni correction for multiple comparisons (α < 0.008, Table 2, Appendix A).

We then ranked microstate parameters, which indicated that the features from microstates C and D had the highest predictor importance scores (Figure 3 and Figure 4). Specifically, Ocurrence_C, Coverage_C, MsMC_C, Duration_C, MsMC_B, and Coverage_D were the highest ranked parameters. Furthermore, to compare the performance obtained using ranked microstate features in classifying patients diagnosed with SZ and HC subjects, we implemented models that incrementally considered ranked features (e.g., model 1 included only the Occurrence_C feature, and model 2 considered Occurance_C and Coverage_C as input) The number of features that were fed to the machine learning approach based on the ranked order is presented in Figure 3. Furthermore, the contributions of the value of the feature to the difference between the actual prediction and the mean prediction is estimated as Shapley values in Figure 4. The x-axis indicates the variable name, and the value next to them is the mean SHAP value. On the y-axis is the SHAP value that indicates how much the change in features can positively or negatively affect the probability of prediction.

Classification performance using the optimized machine learning approach relative to the quadratic SVM [27,58] showed that the highest output measure results were obtained when using 19 ranked features as the input of the optimized ML algorithm [ACC = 90.93%, AUC = 0.90%, sensitivity = 91.37%, specificity = 90.48%] (Table 3 and Appendix A, Figure 5, Appendix A). Furthermore, comparable results in terms of accuracy, sensitivity, and ACU were obtained with the optimized ML algorithm by using the first six ranked features. The best-fitted model selected by the optimized algorithm using 19 features was SVM with Gaussian kernel and ‘Ensemble’ when using the 6 most important features, while quadratic SVM was used in the recent study on 19 microstate features.

## 4. Discussion

We employed an optimized machine learning approach to microstate measures and examined their potential for SZ classification. By applying the Bayesian optimized machine learning approach to ranked microstate measures, we were able to discern resting EEG recordings of SZ from HC subjects with high sensitivity, specificity, and accuracy (Table 3, Figure 5 and Appendix A). We also established that with only six features, we could efficiently classify SZ using microstate analyses (Figure 3, Figure 4 and Figure 5 and Appendix A). Overall, findings from this proof-of-concept study show that optimized ML applied to microstate features could contribute to the identification of patients with SZ relative to HC subjects.

In line with previous studies [18,27,28,30,59], we found four microstates (i.e., A, B, C, and D) that had a similar topography in SZ and HC groups. These four microstates explained the global topographic variance and have been suggested to represent distinct functions of the brain [25,27,48]. Specifically, microstates A and B have been associated with the processing of different sensory modalities and with the mental visualization of the situation [60], whereas types D and C have been implicated in attention regulation and default mode functionality, respectively [27]. Although the types and topographies of these microstates were similar between SZ and HC subjects, individual microstate features differed across groups. For instance, the occurrence of microstate B was significantly increased in SZ vs. HC groups, in line with another study that also reported an association of this altered microstate parameter with the positive symptoms of patients with SZ [30]. We also found the mean GFP was significant in SZ vs. HC individuals across all four types of microstates, likely indicating that SZ patients have a higher level of synchronization (i.e., increased cortical power) during resting EEG recordings [61]. The MsCM, a new microstate feature that was calculated for this study showed higher values in HC vs. SZ across types A, B, and C, thus suggesting that GFP topographies are more consistently repeated (i.e., less variability in topographic patterns) in HC relative to SZ patients. Besides MsCM, Type C had reduced occurrence and coverage in SZ vs. HC individuals. These findings are in line with results from a recent meta-analyses of microstate research [18,30]. Of note, microstate C has been linked to the functionality of the saliency network, including the anterior cingulate, inferior frontal gyrus, and insula [20], and aberrant activity in the salience network has been consistently reported in SZ [62,63,64]. Thus, alterations in microstate C parameters further point to dysfunction in this network in SZ. Here, microstate Type D showed an increase in occurrence, duration, and coverage, while previous studies reported a decrease in these parameters [18,27]. Although these discrepancies could be related to methodological differences as well as medication status [15,65], other microstate studies reported an increase in these Type D characteristics, consistent with our findings [30,66]. Microstate D features have been linked to flexible aspects of attention because of their association with the frontoparietal attention network [20]. Studies indicated that impairments of microstates of class D in SZ are associated with deficits in context update, attentional processes, and executive control, which are often observed in these patients [18,67].

Since we observed alterations in most of the microstate parameters computed, we wanted to assess whether some alterations were more relevant than others in differentiating individuals with SZ from HC. We therefore computed the predictor importance score for each of these 20 parameters and found that features from microstate Type C and D were ranked the highest. A reduction in the occurrence and the coverage of microstate Type C were the two top-ranked features, while coverage and duration of microstate D, both of which were increased in SZ vs. HC, were also highly ranked. Given the implication of types C and D in default mode and dorsal attention respectively, these findings suggest that alterations in these domains may be more relevant for SZ classification [20,22]. At the same time, some microstate B features were ranked high as well, including the MsMC that was computed in this study for the first time, and therefore these parameters should be considered and assessed in future studies of SZ. In contrast, Type A microstate features were ranked lower in our study, a finding in agreement with previous work, showing that these features were relatively intact in patients with SZ vs. HC [27].

Multivariate analysis based on machine learning algorithms provides an opportunity to understand the SZ classification by analyzing many features simultaneously. A handful of studies have utilized microstate features and have tested their accuracy and precision to classify SZ with multivariate patterns and have suggested the efficacy of this approach [14,27]. Building on this body of evidence, in this study, we used this machine learning approach to evaluate multiple, ranked microstate features at the same time. As such, we created a more generalized cross-validated model with increased AUC, sensitivity, and specificity. In particular, the optimized machine learning approach employed here was able to achieve greater than 90% efficacy based on the Gaussian SVM using 19 features and greater than 88% accuracy with the ensemble as the best fitted using just 6 features. Importantly, a recent study that used the same dataset for SZ classification reported highest output measures as [Acc = 75.64%,, Sensitivity = 71.93%, and Specificity = 75.50%] with quadratic SVM [27]. Several factors, including the number of EEG trials, the type and number of microstate features (i.e., MsMC), and the feature selection method (i.e., feature importance score calculation) may have contributed to this difference. The accuracy, specificity, and sensitivity scores reported here are comparable with some deep learning approaches that use EEG microstates for SZ classification [68,69,70,71,72,73]. The fact that findings obtained from our optimized ML approach were comparable to deep learning methods in terms of performance but outperformed in computation and time of processing (i.e., our method [order of minutes] vs. [order of hours]) potentially provide a more rapid, efficient way to achieve an optimized SZ classification.

This study has several limitations that should be addressed in future studies. For example, although the number of epochs was large enough for applying the optimized classification approach, the sample size of the SZ and HC groups was rather small; thus, we decided to apply subject-dependent machine learning approach classification on 5-second segments of data. We also chose this approach to make our results more comparable to a previous study using the same dataset [27]. Compared to the subject-dependent method, the subject-independent method may offer greater generalizability regarding learning the HC vs. SZ labels rather than from the individual’s signature. Therefore, future work on larger groups of SZ patients is needed to confirm the findings from this proof-of-concept study on a larger dataset, and also by applying a subject-independent classification. Nonetheless, in this study, we run a leave-one analysis and found that the main findings did not change (Appendix A). Additionally, even though age and gender were matched, obtaining enough data to reflect the broader range of ages in both genders is necessary for the generalization of the trained models. Thus, to increase classification accuracy and develop an accurate model reflecting the general population, the classification performance of microstate characteristics should be examined using data from larger cohorts encompassing the lifespan. This will contribute to establish how specific features of individuals with SZ vs. HC are captured by the machine learning classification method presented here, in line with a personalized medicine approach. Furthermore, in the present study, four microstates were identified which encompassed at least 70% of the global explained variance (GEV) (Appendix A) [14,15,45,49]. While we found that GEV did not significantly change when the number of microstates increased, this could still affect classification performance. Future work should, therefore, also assess whether more than four microstates are identified and whether a different number of archetypes may affect the classification of SZ. Relatedly, in the present study, the characteristics of Type C and D microstates were among the highest ranked features, thus indicating that these parameters may be more reliable diagnostic features than Type A and B features in schizophrenia classification, which eventually could have relevant implication in the day-to-day clinical psychiatry practice [20,22]. Of note, each patient enrolled in this study underwent a medication washout period of at least seven days before the EEG recordings were performed. Nonetheless, future work should confirm these findings in medication-naïve patients and/or more thoroughly assess the possible impact of antipsychotic medications on EEG microstate parameters.

## 5. Conclusions

By employing for the first time an optimized machine learning approach on microstate measures of resting EEG recordings we achieved higher accuracy, sensitivity, and specificity of SZ patients compared to conventional classification methods, even with just six microstate predictors. Furthermore, our results showed that ranking microstate features was critical to optimize this process. Further studies should confirm and extend these findings on datasets involving larger cohorts of SZ patients. Eventually, the novel machine learning approach employed here may help establish EEG microstates as neurophysiological biomarkers that contribute to the classification of SZ.

## Figures and Tables

**Figure 1 brainsci-12-01497-f001:**
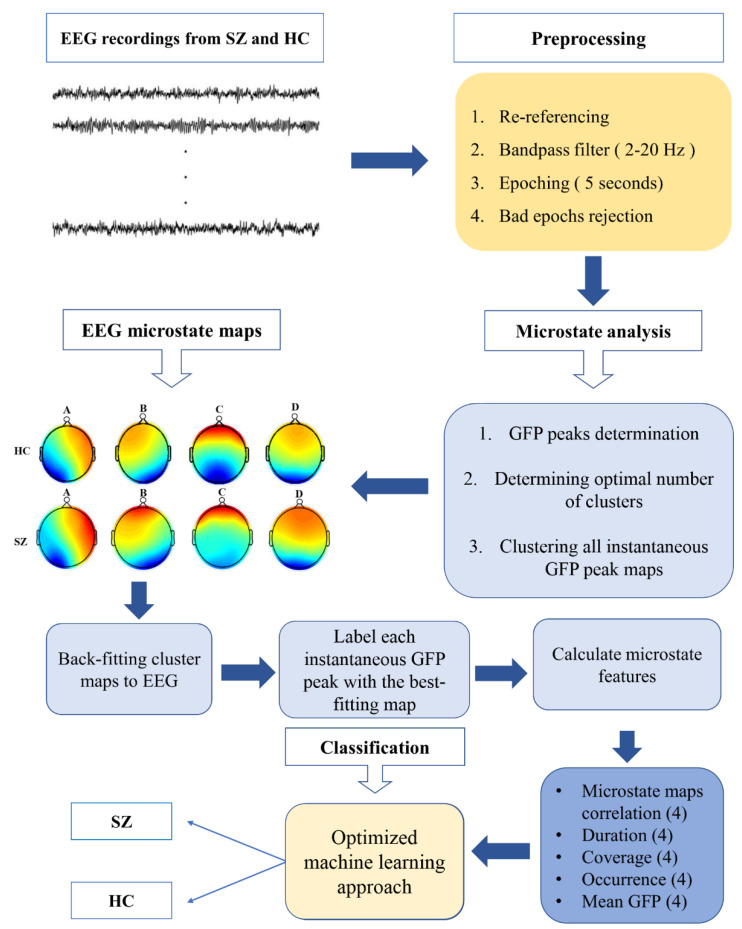
Overview of the process for extracting microstate features; SZ: patients diagnosed with schizophrenia; HC: healthy control subjects.

**Figure 2 brainsci-12-01497-f002:**
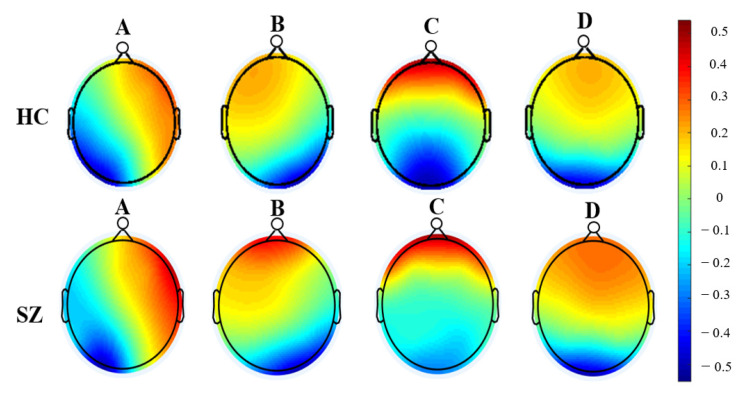
Four normalized microstates (i.e., A, B, C, and D) of resting-state EEG recordings were obtained for patients diagnosed with schizophrenia and healthy control subjects; SZ: patients diagnosed with schizophrenia; HC: healthy control subjects.

**Figure 3 brainsci-12-01497-f003:**
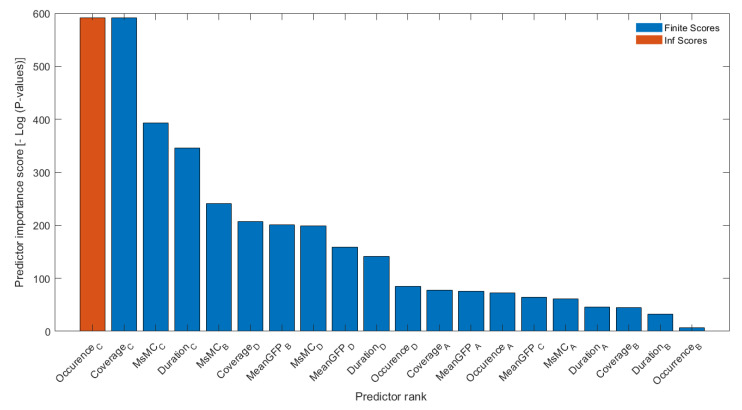
Results of the predictor importance scores for twenty microstate features extracted from resting state EEG recordings of SZ patients and HC subjects.

**Figure 4 brainsci-12-01497-f004:**
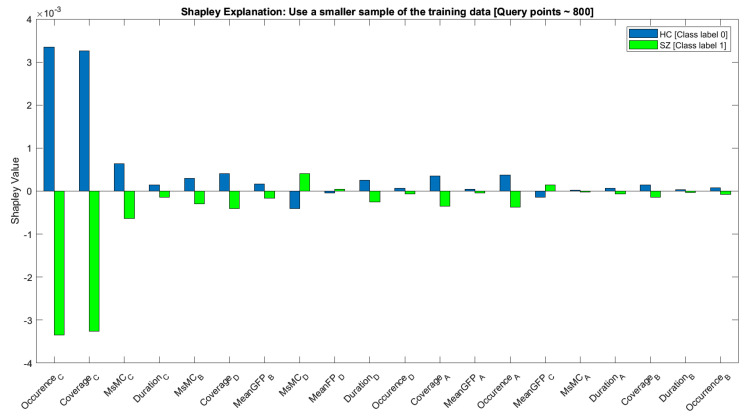
Shapley additive explanation values for the microstate features to explain the contribution of individual features to the prediction.

**Figure 5 brainsci-12-01497-f005:**
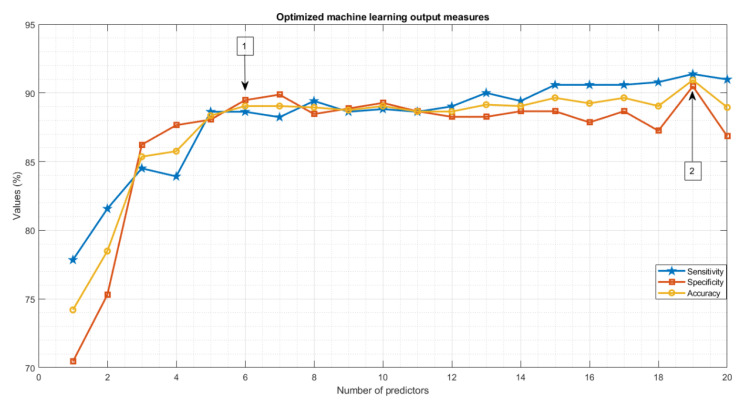
Optimized machine learning results for ranked twenty microstate features. Number (1) shows the best-fitted model results acquired for using six ranked features. Number (2) shows the highest output measures when using 19 input features.

**Table 1 brainsci-12-01497-t001:** Description of microstate predictors that were used in this study.

Features	Definition	Number
Occurrence	Occurrence of a microstate per second (Hz)	4
Duration	The average duration of a microstate (ms)	4
Coverage	Percent of time occupied by a microstate (%)	4
Mean GFP	Mean of global field power (uV)	4
Microstate map correlation (MsMC)	Spatial correlation between topographies and microstates	4

**Table 2 brainsci-12-01497-t002:** Univariate analysis results of the microstate predictors.

Features (Occurrence)	Type A	Type B	Type C	Type D
Mean ± SD [HC]	3.65 ± 1.17	4.26 ± 1.39	4.13 ± 1.28	4.01 ± 1.26
Mean ± SD [SZ]	4.05 ± 1.23	4.41 ± 1.06	2.34 ± 1.42	4.49 ± 1.09
*p*-value	**<0.001**	**<0.001**	**<0.001**	**<0.001**
*t*-value	−11.68	−4.28	46.89	−14.41
Observed power	>0.999	0.990	>0.999	>0.999
**Features (Duration)**	**Type A**	**Type B**	**Type C**	**Type D**
Mean ± SD [HC]	67.18 ± 62.08	58.46 ± 15.80	55.91 ± 15.89	65.07 ± 26.78
Mean ± SD [SZ]	56.22 ± 11.90	59.49 ± 13.94	57.29 ± 36.88	79.79 ± 36.43
*p*-value	**<0.001**	0.015	0.086	**<0.001**
*t*-value	8.68	−2.43	−1.72	−16.30
Observed power	>0.999	0.684	0.405	>0.999
**Features (Coverage)**	**Type A**	**Type B**	**Type C**	**Type D**
Mean ± SD [HC]	23.53 ± 17.62	25.75 ± 10.65	23.65 ± 10.49	27.04 ± 13.36
Mean ± SD [SZ]	22.82 ± 8.31	26.22 ± 8.18	15.21 ± 14.37	35.73 ± 15.507
*p*-value	0.068	0.077	**<0.001**	**<0.001**
*t*-value	1.82	−1.76	23.76	−21.28
Observed power	0.446	0.423	>0.999	>0.999
**Features (Mean GFP)**	**Type A**	**Type B**	**Type C**	**Type D**
Mean ± SD [HC]	4.61 ± 1.67	4.92 ± 1.75	5.20 ± 2.00	5.40 ± 1.99
Mean ± SD [SZ]	4.90 ± 3.93	5.05 ± 1.67	5.42 ± 2.75	5.61 ± 1.77
*p*-value	**0.001**	**0.007**	**<0.001**	**<0.001**
*t*-value	−3.29	−2.70	−3.32	−3.85
Observed power	0.908	0.770	0.913	0.971
**Features (Mean MsMC)**	**Type A**	**Type B**	**Type C**	**Type D**
Mean ± SD [HC]	0.60 ± 0.05	0.62 ± 0.11	0.64 ± 0.09	0.66 ± 0.11
Mean ± SD [SZ]	0.58 ± 0.07	0.60 ± 0.08	0.59 ± 0.10	0.66 ± 0.08
*p*-value	**<0.001**	**<0.001**	**<0.001**	0.434
*t*-value	6.86	6.12	15.99	0.78
Observed power	>0.999	>0.999	>0.999	0.1222

Bold values indicate significant results.

**Table 3 brainsci-12-01497-t003:** Classification outputs using ranked microstate features obtained from the EEG dataset for patients diagnosed with SZ and healthy control subjects.

Classifier	Accuracy (%)	AUC	Sensitivity (%)	Specificity (%)	Best-Fitted Model
Previous study [19 features] [27]	75.64	0.80	71.93	75.50	Quadratic SVM
Previous study [15 features] [27]	76.62	-	-	-	Quadratic SVM
Optimized ML [6 features]	89.04	0.89	88.62	89.47	Ensemble
Optimized ML [19 features]	90.93	0.90	91.37	90.48	Gaussian SVM

## Data Availability

Publicly archived datasets analyzed during the study can be accessed via the link: https://repod.icm.edu.pl/dataset.xhtml?persistentId=doi:10.18150/repod.0107441 (accessed on 1 September 2017).

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
