# Peer review of "Bayesian Optimization of Machine Learning Classification of Resting-State EEG Microstates in Schizophrenia: A Proof-of-Concept Preliminary Study Based on Secondary Analysis"

_brainsci, 2022, doi:10.3390/brainsci12111497_

Round 1
Reviewer 1 Report
Keihani et al show a proof of concept study using resting-state EEG-based microstates to classify SCZ vs references adopting a ML and approach with Bayesian optimization.
Overall the reasoning and methods to identify and classify the microstates are state of the art. However, in order to be able to evaluate the study, I need more details on the study design, specifically the splitting of the training and test dataset. This is a major point to outline in the methods section.
The overall cohort is very small, and includes only 14 vs 14 individuals.
When splitting into training and test split, the authors need to make sure that the split is based on an individual level and cannot be random based on the 5seconds intervals used. The ML approach might otherwise be learning the individual's signature rather than the SCZ vs CTRL label.
To ensure generalizability, it would be of utmost importance to keep one set of patients (test-set) totally unseen by the algorithms, which then will run through the feature extraction pipeline followed by the prediction.
As far as I understand from the approach presented, the authors did predict the 2511 cases vs 2511 control epochs and randomly selected a 20% validation set. This could potential lead to overfitting, if the features are different between the few individuals themselves. So the Algorithm wouldn't learn an SCZ feature but the features of the individuals.
The authors need to specify how they addressed this issue in more detail.
Otherwise, the approach is really elegant and novel.
Reviewer 2 Report
dear authors,
some comments to further improve the manuscript.
the title need to be explicit that this is pilot/preliminary work based on secondary data analysis.
the abstract need to be rewritten for clarity and quality of information.
introduction missed some key meta-analyses about eeg findings in sz. I suggest to run an update authors need to establish priors clearly for bayesian approch vs frequentist.
some major limitations in the methods that neend to be addressed are: power analysis, medications (anti-psychotics) I am interested to know particularly if any patient was on clozapine and chlorpromazine equavalent doses. discussion did not touch on this issue that meds can cause eeg abnormalities. finally, mathematically I did not follow training validation approch used by authors can you please add figure.
discussion need to be improved in light of above suggestions. pls add the role of this study to personalized medicine, the day to day clinical psychiatry is not touched.
Minor: English language need to be rechecked because some sentences appear incomplete for ideas.
Reviewer 3 Report
This study aimed to propose a machine-learning approach to classify resting-state EEG microstates in schizophrenia. I have the following suggestions.
- What is the novelty of this study although several a machine learning approach has been proposed earlier to classify resting-state EEG microstates in schizophrenia?
2. Authors need to add a table including the data structure, details of data source, and size of dataset.
3. Data processing should be more detailed and signal specific. As example, EEG is highly sensitive to the powerline, muscular and cardiac artifacts. In EEG data preprocessing, authors need to mention how you handle AC power, ECG, and EMG artifacts in EEG signals. Same for EOG, EMG and others. Do the authors think that their proposed method is robust to such kinds of artifacts?
- Authors should improve conceptual figures of their ML proposed frameworks with more details and model parametrization.
- Which feature selection method authors used in this study?
- It is recommended to report SHAP (Shapley Additive explanations) values for feature importance.
- Authors should introduce the biosignal applications in ML-based disease prediction in broad scope, such as article, healthsos: real-time health monitoring system for stroke prognostics; and in article, big-ecg: cardiographic predictive cyber-physical system for stroke management, and in article, quantitative evaluation of task-induced neurological outcome after stroke.
- The authors need to mention the model parameters or hyperparameters of ML models. The performance of the model is dependent on the selection of the architecture and/or parameters.
- Authors should report more performance measures of classifiers, such as sensitivity, specificity, precision, and negative predictive value from the confusion matrix.
- Both cross-validated training and testing ROC curves of all microstate classes, schizophrenia and healthy control.
- How did the authors deal with dataset class imbalance challenges in ML analysis?
- The discussion section needs to be improved. Authors must make discussion on the advantages and drawbacks of their proposed method with other studies adding a table in the discussion section.
- Clinical explanation of top important features needs to be described in support of reference.
Round 2
Reviewer 1 Report
The authors have now provided an explanation for their study design. However, a subject independent design without proving that the across subject differences are influencing the model is very likely to be not generalizable. this must be adressed much more critically.
One option would to show that the features used for traingin are not different between the indivduals would be to plot the individuals' values (dots) within the violin plot, to provide correaltion matrices as well as descriptive statistics at individual level.
An additional suggestion is to switch to "minimal" subject dependent design, by excluding the data of one individual per group and in addition to the current appraoch, test the model on these two individuals. This would defintiley strengthen the approach and show its clinical validaty and generalizability
Minor comments:
Hyperparamters of the final Model should be shown, Hyperaparatmers (ranges) used for tuning the models should be provided
Data showing SCZ vs HC should be plotted within one plot or at least have the same scales. With this small sample size showing the individual data points is adviced.
Reviewer 2 Report
thank you for addressing my concerns
Reviewer 3 Report
Few of my comments are addressed. Still few are missing:
Figure S2 doesn't contailns the values of hyperparameters.
Authors need to cite SHAP references and a few details in the methodology.
It is suggested to add a few more details about recommended references in the last comments.
ROC Curves of each microstates nee to be reported.
Which SHAP plot did the author use in this study? The standard SHAP plot looks different in visualization.
